# Piezoelectric Electrospun Fibrous Scaffolds for Bone, Articular Cartilage and Osteochondral Tissue Engineering

**DOI:** 10.3390/ijms23062907

**Published:** 2022-03-08

**Authors:** Frederico Barbosa, Frederico Castelo Ferreira, João Carlos Silva

**Affiliations:** 1iBB—Institute for Bioengineering and Biosciences, Department of Bioengineering, Instituto Superior Técnico, Universidade de Lisboa, Av. Rovisco Pais, 1049-001 Lisboa, Portugal; frederico.porto@tecnico.ulisboa.pt (F.B.); frederico.ferreira@tecnico.ulisboa.pt (F.C.F.); 2Associate Laboratory i4HB—Institute for Health and Bioeconomy, Instituto Superior Técnico, Universidade de Lisboa, Av. Rovisco Pais, 1049-001 Lisboa, Portugal; 3CDRSP—Centre for Rapid and Sustainable Product Development, Polytechnic of Leiria, Rua de Portugal-Zona Industrial, 2430-028 Marinha Grande, Portugal

**Keywords:** articular cartilage, bone, electrospinning, piezoelectric fibrous scaffolds, osteochondral tissue, tissue engineering

## Abstract

Osteochondral tissue (OCT) related diseases, particularly osteoarthritis, number among the most prevalent in the adult population worldwide. However, no satisfactory clinical treatments have been developed to date to resolve this unmet medical issue. Osteochondral tissue engineering (OCTE) strategies involving the fabrication of OCT-mimicking scaffold structures capable of replacing damaged tissue and promoting its regeneration are currently under development. While the piezoelectric properties of the OCT have been extensively reported in different studies, they keep being neglected in the design of novel OCT scaffolds, which focus primarily on the tissue’s structural and mechanical properties. Given the promising potential of piezoelectric electrospun scaffolds capable of both recapitulating the piezoelectric nature of the tissue’s fibrous ECM and of providing a platform for electrical and mechanical stimulation to promote the regeneration of damaged OCT, the present review aims to examine the current state of the art of these electroactive smart scaffolds in OCTE strategies. A summary of the piezoelectric properties of the different regions of the OCT and an overview of the main piezoelectric biomaterials applied in OCTE applications are presented. Some recent examples of piezoelectric electrospun scaffolds developed for potentially replacing damaged OCT as well as for the bone or articular cartilage segments of this interfacial tissue are summarized. Finally, the current challenges and future perspectives concerning the use of piezoelectric electrospun scaffolds in OCT regeneration are discussed.

## 1. Introduction

A material is considered piezoelectric when it can convert an applied mechanical stimulus into an internal electrical signal, with the opposite process also being possible (reverse piezoelectric effect) [1]. The mechanisms that rule this phenomenon are different depending on the nature of the material. For organic materials, this process is related to the reorientation of the molecular dipoles inside the bulk polymer structure (polarization), which is promoted by either the stretching of the material or the application of an electrical field. This leads to the formation of a high net dipole moment that, in turn, induces an electrical stimulus [2]. For inorganic materials, the piezoelectric phenomenon is associated with the displacement of ions inside the crystalline structure [2]. This displacement occurs when the crystal is placed under mechanical stress, leading to changes in the atomic structure of the material and, therefore, to shifts in the balance of ions in the structure and the creation of a dipole moment. Most inorganic piezoelectric materials have a non-centrosymmetric atomic structure that ensures the formed dipoles are not canceled by other dipoles created in the unit cell due to mechanical loading, thus allowing a net polarization to develop [2]. Some centrosymmetric inorganic materials experience breaks of symmetry at nanoscale dimensions or in nonequilibrium conditions enabling them to develop a net polarization, unlike other centrosymmetric materials, and to become piezoelectric [2,3].

The piezoelectric effect is described by four piezoelectric coefficients: *d_ij_*, *e_ij_*, *g_ij_* and *h_ij_*, which correlate the mechanical stimuli applied to the piezoelectric material with the generated electrical stimuli (direct piezoelectric effect) or vice-versa (reverse piezoelectric effect). These coefficients are related via different linear elastic and dielectric parameters [4]. The piezoelectric charge coefficient (d), is the most commonly used as it refers to the polarization generated per unit of mechanical stress, measured in Coulomb/Newton, or, alternatively, to the piezoelectric material’s mechanical strain generated per unit of electric field applied, measured in Meter/Volt (these units are interchangeable) [5,6]. This coefficient is computed using the following relations:(1)$$d_{ij}={\left(\frac{\partial D_{i}}{\partial X_{j}}\right)}^{E}\left[C/N\right]\;$$
(2)$$d_{ij}={\left(\frac{\partial x_{i}}{\partial E_{j}}\right)}^{X}\left[m/V\right]\;$$
where *D* is the electric induction, *X* is the mechanical stress, *E* is the electric field strength, *x* is the strain, and *i* = 1 – 3 and *j* = 1 − 6 [6]. Similarly to the other coefficients, the piezoelectric charge coefficient can be computed by two different expressions, with the first one being relative to the direct piezoelectric effect (charge displacement, *D*, generated by mechanical stimuli, *X*) (1) and the second one being correlated with the reverse piezoelectric effect (mechanical strain, *x*, resulting from applied electrical stimuli, *E*) (2) [4].

Almost all biological matter exhibits piezoelectric properties. This is due in part to the common non-centrosymmetric structure of polar group-containing macromolecules (often helical in shape) that comprise the extracellular matrix (ECM) components of native tissues, such as keratin, which enable the generation of high net dipole moments as a result of mechanical loading [7,8]. Tissues such as bone, cartilage, tendon, skin, ligament, and hair are some examples of piezoelectric tissues whose piezoelectricity stems from the presence of piezoelectric proteins [7]. Overall, these electrical properties have a meaningful impact on tissue function, being directly involved in tissue regeneration mechanisms, and being responsible for modulating cell behavior (bioelectricity) [9,10]. Although the mechanisms through which endogenous or exogenous electrical stimulation of human tissues (e.g., bone) affect cell function are not fully known given their apparent overarching effect, some processes have already been identified. The most prominent effect is related to the destabilization of the resting membrane potential of the cells, which is responsible for triggering the opening of voltage-gated calcium channels (VGCC) and, therefore, promoting calcium intake by the cells [11]. The increase in intracellular calcium levels is, in turn, responsible for triggering the intracellular calcineurin and calmodulin pathways which modify the gene expression profile of these cells and promote the production of growth factors (e.g., TGF-β, BMP-2) [11]. Other potential mechanisms through which electrical stimulation modulates cell behavior may include the activation of mitogen-activated protein kinase (MAPK) signaling pathways, the conversion of said electrical stimuli into mechanical signals (inverse mechano-transduction), which promote actin rearrangement, and the reorganization of the cytoskeleton as well as the increase in ATP production [11,12,13].

The osteochondral tissue (OCT), a complex biological structure placed at the end of long bones and comprised of both articular cartilage (AC) and subchondral bone (SB), is a piezoelectric tissue with important load-bearing functions [9]. These piezoelectric properties, which stem from the presence of collagenous proteins, will be discussed in more detail in the following section. As a result of mechanical wear, aging, and other factors, the OCT is commonly subject to gradual deterioration and to the formation of lesions, which have a meaningful impact on the quality of life of the patients [14]. Osteoarthritis (OA), which is related to the gradual breakdown of the cartilage and bone regions of the OCT due to mechanical wear of the joints, is one of the most prevalent diseases among the adult population worldwide (affecting 303 million people as of 2017) with its prevalence expected to increase over the coming years [15,16,17]. Since the OCT is incapable of self-healing properly due to AC’s avascular nature, clinical treatments are often necessary in case of lesion or disease [18]. Current clinical treatments, including microfractures, allografts, and autografts, have failed to properly address this issue given their concerning lack of reproducibility, poor long-term performance, and often damaging outcomes (e.g., infection, graft vs. host disease) [14].

For this reason, osteochondral tissue engineering (OCTE) strategies involving the development of scaffolds capable of replacing damaged OCT and promoting its regeneration have gained interest recently. Despite the potential of developing piezoelectric scaffolds capable of emulating the native piezoelectric properties of the OCT and providing a platform for in vivo electrical stimulation to enhance tissue regeneration (without the need for external stimulation sources/electrodes), most OCTE studies tend to overlook this strategy, focusing primarily on mimicking the main structural and mechanical features of the tissue.

The OCT’s ECM is characterized by a high percentage of collagenous fibers in its composition: while type II collagen fibers can be primarily found in the AC region of the OCT, the osseous layers of the OCT are mainly comprised of type I collagen [19,20,21]. As a result, this tissue’s ECM has a fibrous structure, which should be replicated when developing scaffolds for OC regeneration. Electrospinning, a technique used to generate fibers with small diameters, ranging from a few nanometers to micrometers, has been applied to develop fibrous scaffolds for OCTE strategies [22,23,24,25]. These scaffolds have enhanced tissue regeneration by providing an improved platform for cell attachment and growth and more native-like biomechanical cues (compared with other scaffold fabrication techniques) [24,26]. Combining the electrospinning technique with piezoelectric biomaterials has the potential to produce advanced bioscaffolds capable of mimicking the structural and piezoelectric properties of native OCT ECM and of being used as physiologically relevant platforms for the electrical or mechanical stimulation of cells, aiming to improve their differentiation towards OCT-related lineages in vitro and enhance OCT regeneration in vivo (Figure 1).

In this review, a summary of the piezoelectric properties of the different regions of the OCT and an analysis of the main piezoelectric biomaterials applied in OCTE strategies are presented. Examples of some of the piezoelectric electrospun scaffolds that have already been developed for potentially replacing damaged OCT, bone, and articular cartilage, as well as some current challenges and future perspectives, will also be discussed.

## 2. Piezoelectric Properties of Bone and Articular Cartilage

Given its heterogenous constitution, the OCT is characterized by multiple continuous gradients present within its structure that shift between its cartilaginous and osseous layers. The difference in ECM composition between AC and the underlying SB is responsible for differences in the electrical properties of both tissues (Table 1) [9].

### 2.1. Articular Cartilage

The electrical properties of the AC layer of the OCT are primarily related to the presence of negatively charged carboxyl and sulfate groups attached to the glycosaminoglycans (GAGs) hyaluronan, and keratan sulfate/chondroitin sulfate, respectively, with the tissue presenting a fixed charge density (amount of fixed charges per volume of intratissue water) ranging from 0.04 to 0.2 mEq/mL [9,35,36,37]. These fixed charges create streaming, diffusion, and *Donnan* potentials [35,36,37].

The cartilage tissue is also piezoelectric, with a piezoelectric charge coefficient (d_33_) reported being between 0.2–0.7 pC/N due to the presence of type II collagen fibers in its structure [9,33,34]. These macromolecules, mainly constituted by glycine, proline, and hydroxyproline residues (with CO and NH units), reorient their dipole moment towards the long axis of the protein once the fibers are placed under mechanical stress. A change in the magnitude of the dipole moment also occurs [2,33,38]. Together these effects are responsible for the piezoelectricity of collagen. The piezoelectricity of AC is very important for the function of this connective tissue. When AC is under mechanical stress, internal electrical signals are produced by type II collagen fibers which trigger ECM production, cell growth, and tissue regeneration, thus optimizing the mechanical response of the tissue [39].

### 2.2. Bone

The bone tissue is generally considered a dielectric material, i.e., a material with reduced conductivity that, in the presence or absence of an external electric field, is capable of generating dipoles through the residual mobilization or separation of negative and positive charges (electric polarization) [10,40]. Charge displacement in the bone tissue, and thus its dielectric properties, are related to the hydrogen bonds present in collagen and hydroxyapatite (HAp) [10,41]. Given its high degree of anisotropy, the conductivity of the tissue depends on the direction of the electrical flow through the bone [9].

Bone is also piezoelectric due to a significant presence of type I collagen fibers in its ECM composition that, as previously stated, are piezoelectric: the piezoelectric constant (d_33_) of biological bone has been reported to be between 0.7–2.3 pC/N [33,42]. Compared with cartilage, bone has a higher piezoelectric coefficient (approximately 28–32% higher), likely due to differences in the molecular structure of the different types of collagen and differences in crosslinking, dielectric properties, and the sequence of amino acid residues [9,43]. Recently, few studies have also found that HAp, an important component of the inorganic phase of bone ECM, was also responsible for bone piezoelectricity, despite initial reports that this phenomenon was solely related to the presence of the collagen fibers [3]. HAp is also responsible for the bone tissue’s pyroelectricity: uniform heating or cooling causes changes in the polarity and electric charges of the connective tissue [3,9,44].

Similarly to AC, the piezoelectricity of bone tissue is very relevant for its function, being often considered one of the main mechanisms behind *Wolff*’s Law, i.e., loads applied to the bone lead to the formation of internal electrical signals produced by type I collagen fibers (as a result of their piezoelectricity) which in turn attract osteogenic cells, triggering bone remodeling and tissue formation, optimizing its mechanical response to deformation [45].

## 3. Piezoelectric Materials

As previously stated, the use of piezoelectric materials for scaffold-based OCTE applications has gained interest recently because, besides mimicking the piezoelectric nature of tissues, piezoelectric biomaterials can also generate relevant physical cues that promote tissue repair and regeneration without the necessity for electrodes, which are often responsible for triggering an adverse chronic foreign body response by the patient’s immune system leading to the formation of fibrotic capsules surrounding the devices and inevitably hindering their effectiveness [1,33,42,46,47]. Optimized piezoelectric scaffolds can produce similar electrical signals to those generated by the native tissue’s ECM when placed under mechanical stress [42]. Piezoelectric materials are frequently poled, i.e., a strong electric field is applied to the materials at usually high temperatures (below the material’s Curie point) to increase charge separation and polarization to improve their piezoelectricity [5]. The two most common piezoelectric materials applied in tissue engineering (TE) are piezoceramics and piezoelectric polymers [10,33,34,42].

### 3.1. Piezoceramics

Piezoceramics, which include titanates (composed of titanium oxides), lead-based ceramics, and lead-free ceramics such as barium titanate (${\mathrm{BaTiO}}_{3}$), lead zirconate titanate (PZT), and zinc oxide (ZnO) are characterized by their excellent piezoelectricity as well as high hardness, making them particularly interesting for mimicking the features of hard tissues, such as bone (Table 2) [33,34,42]. These ceramics are often incorporated with other non-piezoelectric materials (e.g., polymers), in the form of nanoparticles or powders, as piezoelectric additives [33,42,48]. Even though many types of piezoceramics exist, only a few are suitable for TE as most of these materials, especially the lead-based ceramics, are notorious for their toxicity and cytotoxicity, even at low doses, which ultimately restricts their applications as biomaterials [33,49]. The application of piezoceramics in TE is also restricted by the high temperatures often required to perform poling/polarization (usually >600 °C) [50]. Barium titanate as well as HAp, boron nitride (BN), potassium sodium niobate (KNN), and ZnO are among the most commonly used bioactive piezoceramics, having found applications in OCTE and both bone and cartilage TE (with some examples provided in Section 4), as well as more generally within the TE research field (Figure 2) [33,42].

Barium titanate, the first lead-free piezoceramic to be developed and the most studied piezoceramic in TE, is highly biocompatible, with several in vitro and in vivo studies reporting their ability to promote cell attachment, cell proliferation, and tissue growth, among other relevant cellular activities [33,42,54]. Multiple studies have also identified the osteoconductive and osteoinductive potential of this bioactive ceramic [42,48,58]. Barium titanate is also characterized by a high Young’s modulus (118 GPa) and compressive strength (913.2 MPa) [59]. When mechanical stress is applied to this piezoceramic, the titanium ($T_{4}{}^{+}$) and oxygen ($O_{2}{}^{-}$) ions switch positions relative to each other within the ceramic’s perovskite structure, leading to the accumulation of negative charges on the surface and concentration of positive charges inside the crystal (formation of dipole moment) [33]. As a result, when incorporated in vivo within a load-bearing tissue (such as the OCT-, cartilage- or bone ECM), being subject to native loads produced by skeletal mobility, this material is capable of attracting different cations, such as ${\mathrm{Ca}}^{2+}$, from biological fluids enabling the formation of an apatite-like structure on its surface, which improves its osteoconductive potential and its integration in the host tissue [33]. Barium titanate’s non-biodegradability coupled with its brittle nature, poor thermal stability, and reports of cytotoxicity (albeit low) constitute important limitations to using this biomaterial. Nevertheless, barium titanate has thus far been mostly applied in bone TE applications as a piezoelectric additive and has, as of recent, been frequently combined with HAp, with multiple studies reporting improved bioactivity and osteogenic capabilities of the composite [33,48,49].

HAp is one of the most common bone artificial substitutes, being mainly used due to its excellent biological properties such as its high biocompatibility, bioactivity, and biodegradability, as well as its osteoconductive potential and osteointegration capacity [49]. Moreover, HAp is also inherently piezoelectric, with an estimated piezoelectric coefficient in the range of 1.5 to 2.4 pC/N [33,49]. Several studies have taken advantage of this electrical property by poling/polarizing the crystalline structure of HAp, enabling significant increases in the piezoelectricity of the material (repositioning of $H^{+}$ and ${\mathrm{OH}}^{-}$ dipoles). Polarized HAp has been shown to significantly improve cell adhesion and proliferation, as well as to accelerate the formation of a mineralized matrix and ossification [33,49]. The main obstacle to the polarization of HAp is the high temperature (between 300 and 500 °C) and electrical voltage required to achieve sufficient charge separation in HAp-based ceramics [33]. HAp crystals are also characterized by a lower elastic modulus (54–79 MPa) in comparison to the other piezoceramics and low tensile strength (38–48 MPa), which has also hindered its applications [60,61].

ZnO nanoparticles have found many applications in biomedicine because of their biocompatibility, antimicrobial activity, and good chemical and physical properties, exhibiting a Young’s modulus of 140 GPa [49,62]. Due to the critical role of Zn in bone development, this ceramic has an interesting potential in bone TE [33,42]. Additionally, ZnO has a reasonably high piezoelectric coefficient and has been shown to improve the mechanical and piezoelectric properties of various scaffolds (including HAp-based scaffolds) [48,49]. Similar to barium titanate, the cytotoxicity of ZnO (especially in nanometer-sized particles) has thus far hindered its application in TE [34].

Boron nitride nanotubes (BNNTs) characterized by their high piezoelectric coefficient (though smaller than other piezoceramics), biocompatibility, good mechanical properties (with an elastic modulus of 865 GPa and a tensile strength of 33 GPa), and large surface area have also been explored for bone TE applications [42,49,63,64]. BNNTs are also notorious for being easily internalized by cells, allowing them to modulate cell behavior by generating intracellular electrical stimuli [42]. While this piezoceramic has been found to promote the proliferation, attachment, and osteogenic differentiation of MSCs, several studies have also reported their negative influence over chondrocytes, fibroblasts, and smooth muscle cells [42,48]. Contradictory data have been published regarding the toxicity and cytotoxicity of BNNTs, with the potential adverse health effects of this bioceramic restricting their current applications in TE [65].

KNN, an eco-friendly high-performance piezoceramic, is characterized by its high piezoelectric coefficient, good biocompatibility, and thermal stability [66]. This material has also been found to have remarkable antibacterial effects and low cytotoxicity, with some studies reporting the piezoceramic’s ability to promote protein adsorption and cell proliferation [33,66]. KNN is also characterized by a high Young’s modulus (104 GPa) [67]. In multiple instances, lithium ions (${\mathrm{Li}}^{+}$) have been used to dope KNN (LKNN), improving the material’s piezoelectricity, biocompatibility, and chemical stability [10,33,42]. Lithium niobate crystals have been found to enhance osteoblastic proliferation and activity, resulting in accelerated bone regeneration [33,42,57,68]. However, the release of ${\mathrm{Li}}^{+}$ ions has also been related to increased cytotoxicity [33,42]. Few studies have described the development of both polarized and unpolarized KNN and LKNN-based implantable scaffolds for bone, neural, and skin regeneration, with promising results being reported, especially for the polarized scaffolds given the meaningful role of polarized surfaces in modulating cell function as well as other biological responses, such as protein adsorption [10,33]. Additionally, other studies have highlighted the use of this piezoceramic as a drug carrier device capable of modulating drug release through its piezoelectric features: the voltage generated through the mechanical stimulation of the device in vivo can initiate and regulate drug release patterns. However, despite its promising potential applications within TE, research surrounding the use of either KNN or LKNN is still quite rare [10,33,34,42].

The advantages and disadvantages associated with the use of each piezoceramic discussed in this section are summarized in Table 3.

### 3.2. Piezoelectric Polymers

Piezoelectric polymers (PZPs) provide a more flexible platform for producing piezoelectric scaffolds with variable physical, mechanical and chemical properties that can be easily tuned for each application, merging the simple processing of polymers with excellent intrinsic piezoelectric features (Table 4) [10,42,73,74]. Compared with piezoceramics, PZPs are usually not cytotoxic and have improved mechanical properties, including higher strength, high impact resistance, and an enhanced ability to deform, which allows them to be applied not only to mimic the features of hard tissues but also soft tissues [5,10,42,45,49,74,75]. PZPs also provide a more environmentally responsible alternative to producing piezoelectric scaffolds when compared with piezoceramics (specially lead-based ceramics) and are usually easier to be polarized as they have significantly lower Curie temperatures [33,74]. Multiple studies have reported the crucial role of these piezoelectric biomaterials in promoting cell proliferation, differentiation, and other relevant cellular activities, in both in vitro and in vivo settings, with some studies, in particular, demonstrating their osteogenic capacity [48,74]. PZPs, which have found many applications in bone, neural and cardiac tissue regeneration, have been typically fabricated as fibers, films, or rods [6,49]. Poly(vinylidene fluoride) (PVDF) and its derivative Poly(vinylidene fluoride-trifluoroethylene) (PVDF-TrFE) as well as poly-l-lactic acid (PLLA) and the natural polymers poly(3-hydroxybutyrate) (PHB), collagen, cellulose, and chitosan, are the most commonly used PZPs, having been applied in OCTE and both bone and cartilage TE (with some examples provided in Section 4), as well as more generally within the TE research field (Figure 3) [73].

PVDF is a semicrystalline polymer (50–60% crystalline fraction) constituted by two polar fluorine domains attached to every second carbon atom along a vinyl backbone [5,34,75]. This PZP is comprised of five crystalline phases (with different dipole orientations) that greatly influence its properties: α, β, γ, δ and ϵ, with the α and β phases being the most common (the γ phase is a transition state between the α and β phases, while the δ and ϵ phases are rarely observed through conventional processing techniques) [5,75]. While in the α phase (the most thermodynamically favorable), the polymeric chains are packed in such a way that the molecular dipoles become antiparallel (trans-gauche conformation), creating a non-polar and non-piezoelectric crystal structure, in the β phase, the polar fluorine groups are parallel (all-trans conformation) thus generating a high net dipole moment and contributing to the polymer’s piezoelectricity (Figure 4) [1,2,5,45]. Since PVDF is mostly settled in its α phase (α-PVDF), different strategies, including polarization, stretching, annealing, and the addition of nanoparticles (e.g., graphene oxide, carbon nanotubes, graphene), have been developed to increase the β phase content of PVDF to improve its piezoelectricity [10,33,74,75,78]. Depending on the dipole arrangement, the surface of PVDF becomes charged with either a positive or negative net charge which affects its interaction with cells. Negative charges promote cell adhesion as well as the adhesion of proteins and cations, improving the biocompatibility of the synthetic polymer [1,73,75]. The high piezoelectric coefficient of β-PVDF coupled with its flexibility, ease of processing, high chemical and physical resistance, and non-toxicity have made this polymer one of the most commonly used PZPs in TE applications [33,42,45,49]. This biocompatible thermoplastic has been found to improve cell adhesion and proliferation to different extents depending on poling, degree of hydrophilicity/hydrophobicity, and scaffold morphology. In particular, multiple studies have identified the osteogenic potential of PVDF and the importance of the piezoelectric signals generated by the polarized surfaces in augmenting MSC differentiation and bone growth [10,33,42].

Poly(vinylidene fluoride-co-tetrafluoroethylene) (PVDF-TrFE) is a copolymer of vinylidene fluoride (VDF) and trifluoroethylene (TrFE), which adds three extra polar fluorine domains to every other carbon atom along the polymeric vinyl backbone. PVDF-TrFE is characterized by an intrinsically high content of β phase, which is more thermodynamically favorable when compared with PVDF due to the steric hindrance stabilization (molecular congestion caused by the physical presence of bulky surrounding ligands) generated by the TrFE residues [1,5,33,42,75]. As a result, stretching, annealing, or poling of PVDF-TrFE are not required to increase its β phase content and, therefore, make the material piezoelectric, although such post-processing techniques can be used to enhance its piezoelectricity [74,75] further. In fact, due to its lower Curie temperature compared to PVDF (PVDF: 170 °C; PVDF-TrFE: 65 °C), PVDF-TrFE is easier to polarize [33]. Depending on the ratio of VDF and TrFE, the β phase content of the polymer can be adjusted to more appropriately fit different applications and elicit different cellular responses [33]. Overall, PVDF-TrFE has a high piezoelectric coefficient (the highest among organic PZPs). Similar to PVDF, it is biocompatible, having a positive influence over cell adhesion and proliferation [42,49,79]. The chondrogenic and osteogenic potential of this material have been previously reported in different studies [49,80]. This PZP has been used to mimic different piezoelectric tissues, including the OCT, skin, tendon, and neural tissues [42,49]. PVDF and PVDF-TrFE have also been applied as a coating for bone implants to enhance osteogenesis and improve integration with the host tissue [48,81]. The main limitation associated with the use of PVDF and its derivatives is its non-biodegradability, which restricts its clinical application as temporary tissue implants [10,33,42,48]. It should be noted that PVDF is among the few ferroelectric materials with negative piezoelectric charge coefficients, which means that it exhibits inverse strain-field and polarization-stress characteristics. When exposed to an electric field which would make most piezoelectric materials expand, PVDF is compressed [2].

The polyesters PLLA and PHB have garnered significant attention due to their piezoelectric features and good biodegradability [1,33,34,45,48]. PLLA, the most common form of PLA, is a semicrystalline polymer with a helical structure that has found ample applications in biomedicine [42,73]. As a result of its biocompatibility, non-toxicity, and biodegradability coupled with its adequate mechanical properties, good processability, elastomeric behavior, and corrosion resistance, this FDA-approved polymer is currently used in bone implant devices and temporary surgical fixation devices (e.g., sutures, pins, screws) [1,10,42,45,48,49,73]. Due to the presence of polar carbonyl groups (C=O) in its structure that are displaced when the polymer is placed under mechanical stress (generating a net dipole moment), PLLA is piezoelectric [2,33,49]. This biomaterial constitutes four crystalline phases with its α and β phases being the most relevant: in α-PLLA (the most thermodynamically favorable), the dipoles are randomly oriented, while in β-PLLA, the carbonyl groups are aligned [1,2,33]. Even though non-poled α-PLLA is naturally piezoelectric, mechanical stretching and poling are often used to obtain the more piezoelectric β-PLLA conformation [42]. While poled PLLA has been shown to enhance bone formation in vivo, further studies are required to complement the currently insufficient data regarding the effects of this poled PZP on the differentiation and proliferation of bone-related cells [1,10,33,45,48,49,82].

PHB, a natural polymer widely found in the cytoplasm of prokaryotic cells (especially bacteria), is also piezoelectric (with a low piezoelectric coefficient when compared with other PZPs) [1,73]. Similar to PLLA, the piezoelectricity of PHB is related to the presence of polar carbonyl groups in its multi-lamellar crystalline structure [33,73]. Both poled PHB and its copolymer PHBV are piezoelectric and are starting to be applied in TE, particularly bone TE, because of their attractive properties, which include their piezoelectric features as well as their biocompatibility, biodegradability, high stability, good processability, and non-toxicity [1,10,33,42,49,73,83]. Compared with other biodegradable PZPs like PLLA, PHB has the advantage of its long degradation time, which allows it to support the growth of the neo tissue for longer periods [42,49]. However, this material is also characterized by its poor mechanical properties and insolubility in water, which restrict its applications in TE (particularly for load-bearing tissues) [10,49]. Overall, even though PLLA and PHB can be used for developing piezoelectric-based scaffolds for TE, their low piezoelectric coefficients, when compared with PVDF and PVDF-TrFE, have thus far limited their use in piezoelectricity-related applications, with their current use in TE being mostly associated with their advantageous biological features [42,45].

Similarly to HAp, collagen, a major component of OCT ECM, is frequently applied in OCTE applications because of its excellent biological properties such as biocompatibility, biodegradability, low antigenicity, hydrophilicity, and cell-binding ability [10,33,42]. Collagen fibrils can provide relevant biological cues that in turn promote different cellular activities [33]. Additionally, this natural biomaterial is intrinsically piezoelectric due to its non-centrosymmetric structure and to the presence of multiple molecular dipoles [2,10,33]. To improve its piezoelectric properties, hydrolysis-treated collagen (anionic collagen), i.e., collagen with hydrolyzed asparagine and glutamine residues, has been developed as a potential substitute for the bone tissue, with multiple studies reporting its positive effects on osteoblast differentiation as well as in vitro mineralization and alkaline phosphatase (ALP) activity (an important enzyme present in the bone which is often used as a biomarker for bone formation) [10,33,42]. Collagen has also been frequently combined with piezoceramics, including HAp and tricalcium phosphate (TCP), to improve its piezoelectricity [42,84]. However, the use of collagen in TE is heavily hindered by its poor mechanical properties (low stiffness) and rapid degradation [10,42].

Two additional natural biopolymers have also been applied for developing piezoelectric biodegradable scaffolds: cellulose and chitosan. Cellulose, a homopolymer of glucose with excellent biocompatibility, high tensile strength, and the ability to promote significant cell adhesion (particularly with osteocytes, chondrocytes, and endothelial cells), has a small shear piezoelectric charge coefficient (d_31_) of 0.1 pC/N [42,49,76]. This polymer has been frequently applied in many biomedical applications, including bone and cartilage TE and drug delivery, with most of its current applications not being piezoelectricity related [33,49]. A few studies have described the production of piezoelectric HAp/Cellulose composites with promising potential for bone TE. In contrast, other studies have reported the potential of piezoelectric methylcellulose, a cellulose derivative, as an injectable biopolymer capable of repairing brain defects [10,33,76,85,86].

Chitosan, a biodegradable and biocompatible polysaccharide derived from the deacetylation of piezoelectric chitin (0.2–1.5 pC/N) has also been a popular choice in bone and cartilage TE, being characterized by its anti-bacterial activity, minimal foreign body response, and osteoconductivity [10,33]. However, the use of chitosan has been hindered thus far by the material’s poor mechanical properties, which make its clinical application difficult. A few solutions to address this issue have been developed, namely mixing the natural piezoelectric polymer with other synthetic polymers [76]. Overall, despite their potential in TE, both cellulose and chitosan have not attracted much interest for their piezoelectric features [33].

To offset some of the issues related to using the different PZPs, they are often blended with other materials, particularly other piezoelectric materials. Composites constituted by both PZPs and piezoceramics are frequently developed, combining the high piezoelectric coefficient of the latter with the biocompatibility, processability, and flexibility of the former (e.g., PVDF-TrFE/BaTiO_3_). These piezoelectric composites usually have better mechanical stability and mechanical properties than either the PZP or piezoceramic alone [48,73].

The advantages and disadvantages associated with the use of each piezoelectric polymer discussed in this section are summarized in Table 5.

## 4. Applications of Piezoelectric Electrospun Scaffolds in Bone, Articular Cartilage and Osteochondral Tissue Engineering

Overall, the scaffold-based OCTE strategies that have been developed so far can be mostly subdivided into four main categories: (1) monophasic or single-phase scaffolds, that were the first gold standard for OCTE, in which the same homogenous material is used for replicating both the AC and SB layers of the OCT; (2) biphasic scaffolds, the most popular current OCTE strategy, in which two different biomaterials are used to recreate the cartilage and bone phases of the OCT; (3) multiphasic scaffolds (e.g., triphasic scaffolds) in which additional biomaterial layers are added to more accurately replicate the transition phase between the AC and SB layers of the OCT as well as better differentiate the several AC and SB regions; and (4) gradient scaffolds in which a continuous gradient, rather than the discrete gradient used for the other scaffold-based OCTE strategies, is introduced in order to mimic one or more of the several gradients that are found in the native OCT (e.g., mechanical gradient, biochemical gradient) [9,19,88,89,90,91].

Additionally, several studies have combined solid scaffolds for the bone region of the OCT with hydrogels for the AC segment [89,92]. OCT scaffolds have also been combined with multiple bioactive factors (e.g., growth factors, small molecules) that are released in a controlled manner (following a pre-defined pattern) in the patient’s body and promote tissue regeneration and growth as well as the integration of the scaffold with the native tissue [19,26,91,93]. A few studies have also described the development of scaffolds focused solely on the cartilaginous or osseous regions of the OCT.

Different physical and chemical procedures have been developed over the years for processing biomaterials and rendering them useful for OCTE applications via the development of adequately structured scaffolds [94,95]. These strategies include the more conventional techniques such as solvent casting and particulate leaching, freeze-drying, phase separation processes (TIPS-Thermally Induced Phase Separation and DIPS-Diffusion Induced Phase Separation), and gas foaming or more advanced fabrication methods, including rapid prototyping, also known as additive manufacturing, bioprinting and electrospinning [90,94,95,96,97,98]. Unlike the advanced fabrication methods, the traditional scaffold fabrication processes give very limited control over the structure of the scaffolds, providing only minimal control over the scaffold’s degree of porosity and pore size [90].

Additive manufacturing, commonly referred to as 3D printing, is associated with the development of a 3D structure via the layer-by-layer deposition of powder, liquid, or solid material substrates, in a process that is guided by previously developed computer-aided design (CAD) models that are replicated by the 3D printer [90,94,97,98,99]. This versatile scaffold fabrication technique originates scaffolds with good spatial resolution and can generate complex 3D structures in a fast and reproducible manner [94,99]. Bioprinting is another 3D fabrication technique in which bioinks, containing biomaterials and live cells, and, in some instances, bioactive molecules are used to produce tissue analogs using a precise spatial control system [99]. Both 3D printing and bioprinting have been extensively used to manufacture OCTE scaffolds due to their capacity of more closely replicating the heterogenous and complex structure of the OCT when compared with the previously mentioned techniques, coupled with the increased control over the main features of the scaffolds [100]. However, it should be pointed out that these promising strategies present a few limitations, such as some scaffold shape restrictions (due to technical printing issues) and material cytotoxicity, as well as reduced cell viability in the bioprinting processes [97].

As previously mentioned, electrospinning, a versatile, scalable, and relatively simple technique in which electrostatic forces applied to polymeric solutions or molten polymers are used to produce fibers with small diameters (in the micrometer and nanometer range), has also been widely applied in OCTE [22]. Through this scaffold manufacturing strategy, fibrous mats that mimic the fibrous nature of the OCT’s ECM (comprised of a high percentage of collagenous fibers) are produced, providing a platform with nanometer and micrometer-sized features that can potentially enhance the in vivo regeneration of damaged tissue by improving cell attachment and growth, and providing more native-like biomechanical cues [26]. Additionally, due to the small diameter of the fibers obtained, electrospinning also allows the fabrication of scaffolds with high surface-to-volume ratios, flexible surface functionalities, and improved mechanical properties [22,24]. These combined properties make electrospinning a promising strategy for producing scaffolds for OCTE applications.

Electrospinning has also been found to positively affect the piezoelectricity of PZPs by simultaneously applying a high electrical field to the polymeric solution (poling/polarization) and through jet stretching/thinning (mechanical stretching of the polymeric structure). In the particular case of PVDF and PVDF-TrFE, electrospinning has been shown to enhance the material’s piezoelectricity by increasing the polar β phase content (in comparison to the pristine materials) [74,75,79,101,102]. Sengupta et al. detailed the strong dependence of the fraction of β phase content of PVDF with the applied voltage as well as the working distance used during electrospinning [103]. PVDF and PVDF-TrFE fiber alignment has also been found to improve the piezoelectricity of the PZPs [74]. Therefore, besides allowing the production of piezoelectric nanofibrous structures that closely mimic the fibrous nature of different piezoelectric tissues’ ECMs, electrospinning of this type of material also enables the production of high-performance PZPs [75].

The mechanical properties of OCT scaffolds are extremely important given the need for these structures to support the cells and newly formed ECM and to withstand the forces constantly being applied to this load-bearing tissue [26,104]. For this reason, electrospun scaffolds used to replace damaged OCT should exhibit similar mechanical properties to the AC or bone regions of the interfacial tissue. The mechanical properties of fibrous scaffolds are heavily influenced by two factors: (1) the operational parameters used during electrospinning, which enable some optimization of the mechanical features of the fibrous mats; and, most importantly, (2) the intrinsic mechanical properties of the polymers and ceramics used in the formulation of the electrospinning casting solution [105]. Regarding the use of piezoceramics, as previously discussed, these materials are characterized by high Young’s moduli (proportional to their high hardness) which matches more closely the elastic modulus of cortical and trabecular bone (17 GPa and 350 MPa, respectively) [33,34,106]. When incorporated within polymeric fibrous structures as piezoelectric additives (e.g., nanoparticles, nanotubes), piezoceramics have been found to contribute to enhanced mechanical properties. Bagchi et al. described the development of barium titanate-loaded PCL fibers, obtaining piezoelectric fibrous scaffolds with a 34% increase in elastic modulus (424 ± 32 MPa) and a 10% increase in yield strength (15.7 ± 0.7 MPa) in comparison to the unloaded PCL fibers [107]. Similarly, Khader et al. reported an increase of 74% of the elastic modulus of PCL nanofibers (7.3 ± 1.1 MPa) (comparable to the elastic modulus previously reported for cartilage of 12 MPa) through the addition of ZnO nanoparticles (10 wt%) as well as a 67% increase in their ultimate tensile strength (0.65 ± 0.25 MPa) [106,108].

As previously stated, PZPs present higher mechanical flexibility as well as higher strength and lower stiffness compared to piezoceramics, which enable them to be used for replacing not only hard tissues like bone but also soft tissues, such as AC [8,49]. Overall, synthetic PZPs present higher stiffness and strength than natural PZPs [10,33,42]. Multiple PZP-based piezoelectric electrospun scaffolds with a wide range of mechanical properties can be found in the literature, depending not only on the particular PZP used but also on the operational electrospinning parameters used. While some of those polymeric fibers present mechanical properties that would be more appropriate for replacing the AC region of OCT, others present mechanical features more adequate for substituting damaged bone. In a study developed by Damaraju et al., random, deformable, and piezoelectric PVDF-TrFE nanofibers with an elastic modulus of 4.0 ± 0.2 MPa and a high ultimate elongation of 125.17 ± 0.21% were developed for potentially replacing damaged OCT. A ~33% increase of the Young’s modulus of these fibers was achieved through an annealing treatment of the fibrous mats (5.3 ± 2.3 MPa), with a significant decrease in ultimate elongation of the heat-treated fibers being also observed (17.9 ± 2.6%) [80]. In a different study, Sadeghi et al. reported the development of PHB-based piezoelectric fibrous scaffolds with a Young’s modulus of 74.45 ± 2.88 MPa and tensile strength of 87 ± 3 MPa for cartilage TE. By combining another natural PZP, chitosan, they observed, as expected, a decrease in both elastic modulus and tensile strength of the fibrous mats [109]. Baji et al. described the reinforcement of aligned PVDF piezoelectric fibers with barium titanate nanoparticles for bone TE. While the PVDF fibers displayed a Young’s modulus of approximately 1.0 GPa and a tensile strength of 150 MPa, an increase of the stiffness and tensile strength of the fibers by 36% and 12%, respectively, was observed with the addition of the piezoceramic [110].

In the following subsections, examples of piezoelectric fibrous scaffolds developed for bone, cartilage, and osteochondral TE strategies using both lead-free piezoceramics and PZPs will be presented.

### 4.1. Bone

As previously mentioned, research surrounding the use of piezoceramics for developing piezoelectric scaffolds for TE applications has been rather limited due to relevant toxicity and environmental concerns. Nevertheless, few studies have described the fabrication of nanofibers containing lead-free piezoceramics for bone TE.

Busuioc et al. described the development of barium titanate and calcium phosphate containing piezoelectric non-woven fibrous scaffolds for in vivo bone electrical stimulation. To produce these randomly oriented ceramic fibers, template gelatin fibers were first fabricated by blending a fish gelatin precursor solution (70%, *w*/*v*) with double-distilled water. These solutions were then electrospun using a grounded roller collector. After being crosslinked in glutaraldehyde solution, the fibers were loaded with calcium phosphates and, afterward, barium titanate nanoparticles. The resulting piezoelectric scaffolds were then freeze-dried to preserve their 3D structure and heat-treated under different experimental conditions to remove the original gelatin template structure. The authors of this study were able to demonstrate gelatin removal with the increase in temperature and corroborate the presence of both the barium titanate nanoparticles and calcium phosphates on the structure of the fibers using elemental analysis (EDX), X-ray diffraction (XRD) analysis, and Fourier transform infrared spectroscopy (FTIR). Moreover, the versatility of this fiber production method was demonstrated by its ability to tune different morphological features of the fibrous scaffolds (e.g., pore size, shape, and fiber arrangement), which could have an important impact on cell adhesion, proliferation, and differentiation, as well as tissue integration [111].

In a different study, Nagarajan et al. developed boron nitride functionalized gelatin nanofibers to enhance the mechanical properties of the scaffolds (due to their low stiffness) and, therefore, improve their potential use in bone TE applications. The authors started by dissolving gelatin (20%, *w*/*v*) in acetic acid, after which they added boron nitride powder at different concentrations (0.1%, 1%, and 5%, *w*/*v*). Afterward, a conventional electrospinning setup with a rotating drum collector was used to produce the randomly oriented nanofibers. The nanofibers containing boron nitride were found to have increased Young’s modulus and were also stable in aqueous media and biodegradable. These scaffolds were successfully mineralized in simulated body fluid (SBF), a solution with an almost identical ionic composition to blood plasma, suggesting that these scaffolds could potentially be mineralized in an in vivo setting, which constitutes a promising feature for bone TE applications. By seeding the piezoelectric nanofibers with human osteosarcoma cells, it was possible to demonstrate that the scaffolds were not cytotoxic, with cell proliferation profiles similar to the control condition (gelatin only nanofibers). Moreover, the piezoelectric nanofibers were also found to be highly biocompatible and to have an osteogenic potential, as they were able to promote osteoblast gene expression in the absence of any osteogenic growth factors in the cell culture medium [112].

As previously stated, compared with piezoceramics, PZPs have been applied more frequently in TE applications, being used to repair damaged piezoelectric tissues such as bone. PVDF and PVDF-TrFE have been the most commonly used polymers due to their high piezoelectric coefficient and excellent biological and physical properties, while PLLA and PHBV have also been applied to a smaller extent.

Wang et al. described the development of aligned PVDF-TrFE fibrous mats for bone TE, which were poled and annealed afterward (post-processing). PVDF-TrFE (75/25) solutions were prepared with a fixed concentration of 20% (*w*/*v*) using dimethylformamide (DMF)/Acetone (3:2) (*v*/*v*) as solvent system. Using a grounded roller collector during the electrospinning of these solutions, the authors produced aligned PVDF-TrFE nanofibers, which were annealed in a vacuum oven at 135 °C for 4 h. The annealed samples were then pressed in a powder compression machine and placed in a silicone oil bath at 135 °C, where an electrical field of 80–100 MV/m was applied for 30 min (thermal poling treatment). Afterward, the nanofibers were cooled, and constant voltage was applied until they reached room temperature. As expected, the poled and annealed PVDF-TrFE fibers had the largest β phase content (69.2%) compared with the annealed fibers (46.6%) and the non-treated fibers (43.1%) (values computed from XRD patterns). The produced nanofibers were seeded with mouse pre-osteoblasts and were placed in dynamic cell culture plates comprised of a flexible bottom and speakers that emitted low-frequency mechanical vibrations. The results showed that the cells tended to elongate and orient along the direction of alignment of the PVDF-TrFE nanofibers. The authors also verified that the poled and annealed nanofibers induced a considerable increase in osteoblastic cell proliferation compared with the controls, highlighting the potential of PVDF-TrFE fibrous scaffolds for bone TE applications [113].

In order to improve the wettability of PVDF, which is often an obstacle in TE applications, Kitsara et al. described the development of permanently hydrophilic PVDF nanofibers. PVDF solutions were first produced by dispersing PVDF powder at a fixed concentration of 15% (*w*/*v*) in DMF/Acetone (2:3). These solutions were then electrospun using a conventional setup with a static grounded collector, and the resulting fibers were treated with oxygen plasma. As expected, through this surface treatment, the authors were able to significantly enhance the wettability of the fibrous scaffolds, which became very hydrophilic (contact angle decreased from 130° ± 3° to 35° ± 3°). By seeding the processed and non-processed fibers with human osteosarcoma cells, Kitsara and colleagues reported more significant cell spreading and integration for the surface-treated PVDF scaffolds, which led them to conclude that this post-processing strategy could be potentially applied to enhance the integration of β-PVDF scaffolds in vivo and, therefore, improve the regeneration of damaged bone tissue [114].

Tandon et al. developed a PVDF-based piezoelectric composite comprised of PVDF and HAp to improve the osteogenic potential of PVDF fibers. The electrospinning casting solutions were prepared by combining HAp nanoparticles at 5% (wt%) and 10% (wt%) concentrations with DMF/Acetone (1:1) and, after ultrasonication, adding solid PVDF (powder) at a fixed concentration of 16% (wt%). The final polymeric solutions were electrospun using a conventional setup and a rotating drum collector. The random PVDF/HAp fibers showed an increase in average diameter compared with PVDF-only fibers. They were also found to have a reduced β phase content, with the fibers with the highest concentration of HAp (10%) registering the lowest β phase content percentage (77% for PVDF/HAp fibers with 5% HAp, 70% for PVDF/HAp fibers with 10% HAp and 88.02% for PVDF fibers). The addition of HAp was also associated with an increase in the hydrophobicity of the fibers [115].

Using a similar procedure to the one previously described, Gorodzha et al. developed HAp filled PHBV piezoelectric fibrous scaffolds to assess HAp impact on the osteogenic potential of this biodegradable PZP. The authors started by dissolving PHBV (23%, *w*/*v*) in chloroform, after which silicon-containing hydroxyapatite (SiHAp) nanoparticles (10%, wt%) were added. The resulting solutions were electrospun using a rotating drum collector, and randomly oriented fibers were produced. The addition of HAp nanoparticles to PHBV was found to slightly increase the piezoelectric charge coefficient of the fibers (from 0.605 ± 0.093 pC/N to 1.558 ± 0.065 pC/N), which became closer to the values reported in the literature for native bone tissue [30,34]. Cell culture experiments were performed in static conditions using human mesenchymal stem/stromal cells (hMSCs) cultured in an osteogenic medium. Overall, results showed that the functionalized scaffolds presented enhanced cell adhesion and differentiation among all experimental conditions. Moreover, compared with the non-piezoelectric polycaprolactone (PCL) fibers, HAp filled PHBV piezoelectric fibers showed higher calcium accumulation, highlighting the promising potential of these piezoelectric composite fibrous scaffolds for bone TE applications [116].

Another study from Li et al. described the development of barium titanate loaded biodegradable PLLA fibers to enhance the electroactivity of the PZP and, therefore, improve the osteoinductive potential of the fibrous scaffolds. PLLA/Barium Titanate (PLLA/BT) solutions were first prepared by adding surface-modified barium titanate nanoparticles (previously dispersed in sodium citrate) at different contents (1%, 3%, 5%, 7%, and 10%, wt% of the PLLA) to a PLLA solution (in trifluoroethanol). These solutions were then electrospun using a conventional setup with two different collectors: a static grounded collector was used to obtain randomly oriented fibers, while a rotating drum collector was used to obtain aligned fibers. The addition of barium titanate nanoparticles was found to improve the surface roughness and wettability of the fibers, as well as their dielectric properties, with the reported permittivity values obtained in this study being in line with the values reported in the literature for the native bone tissue. The PLLA/BT scaffolds were then seeded with bone marrow MSCs. Results showed improved osteogenic differentiation for the randomly oriented fibers compared to the aligned fibers. A more significant MSC osteogenic activity for the barium titanate loaded fibers compared with the control conditions was also reported, and attributed to the biomimetic electrical activity provided by the PLLA/BT scaffolds [117].

A brief summary of recent studies that have described the production of piezoceramic or PZP-based electrospun nanofibers for bone TE applications, including a more detailed analysis of their main findings, is presented in Table 6.

### 4.2. Articular Cartilage

Research studies involving the design and development of fibrous piezoelectric AC analogs using either piezoceramics or PZPs are quite scarce in the literature, especially when compared with those developed for bone. This is likely due to lesser research focused on the native piezoelectric properties of cartilage compared to bone, which was the first biological tissue where piezoelectric properties were discovered (in the 1970s) [122].

Jacob et al. reported the development of piezoelectric PHBV nanofibers supplemented with barium titanate nanoparticles for the electrical stimulation of cartilage in vivo. The authors started by dissolving PHBV (15% and 20%, *w*/*v*) in Chloroform/Methanol (3:2), after which barium titanate nanoparticles were added at different concentrations (5%, 10% and 20%, wt%). These solutions were electrospun using a conventional electrospinning setup with optimized operational parameters. The addition of barium titanate was found to improve the mechanical properties of the fibers, namely, an increase in Young’s modulus of the PHBV/Barium Titanate scaffolds compared to the PHBV only scaffolds. The piezoceramic was also responsible for increasing the piezoelectric charge coefficient of the fibers, with the composite scaffolds with the highest barium titanate concentration presenting a piezoelectric charge coefficient value close to the one reported in the literature for the native bone tissue (1.4 pC/N). By culturing the functionalized and non-functionalized scaffolds with MSC-derived chondrocytes, the authors observed enhanced cell adhesion, proliferation, and expression of chondrogenic markers (type II collagen) with the increase in barium titanate concentration on the composition of the fibers. The fibers were also electrically poled to improve their piezoelectric features: poled PHBV/Barium Titanate fibers exhibited increased cell proliferation and collagen production compared with the non-processed scaffolds, thus demonstrating the positive effect of piezoelectricity in promoting cartilage regeneration [123].

A different study from Sadeghi et al. reported composite PHB fibers blended with the natural polymer chitosan, which was added to improve the wettability, biodegradability, and biological performance of the piezoelectric scaffolds. PHB/Chitosan solutions were first prepared by adding chitosan at different concentrations (5%, 10%, 15%, and 20%, wt%) to trifluoroacetic acid solutions with fixed PHB concentration (9%, *w*/*v*). A conventional electrospinning setup with a static grounded collector was used for producing the fibers. As expected, an increase in the hydrophilicity of the fibrous scaffolds (contact angle decreased from ~74° to ~67°), as well as an increase in mass loss with time (biodegradability), was reported with the increase in chitosan concentration. PHB fibers containing chitosan were also found to have increased porosity and reduced tensile strength and Young’s modulus. The influence of chitosan presence on cell attachment was tested by seeding the scaffolds with rabbit AC isolated chondrocytes for 4 h before rinsing the scaffolds and fixing the cells with glutaraldehyde: an increase in cell adhesion and spreading was observed via SEM imaging for the PHB scaffolds containing chitosan. Considering both the positive physical and biological attributes of these piezoelectric scaffolds, the authors highlighted the potential of these composite piezoelectric scaffolds for cartilage TE applications [109].

To assess the effect of piezoelectricity and hydrostatic pressure (an important force that exerts loads on cartilaginous cells) on the chondrogenic differentiation of stem cells, Khorshidi et al. described the development of fibrous piezoelectric PVDF/PCL cartilage analogs. The study started by dissolving PCL pellets in dimethylformamide (DMF)/tetrahydrofuran (THF) (1:1) and PVDF powder in dimethyl acetamide (DMAC)/Acetone (1:1). Both solutions were then mixed in a 50:50 ratio and homogenized, after which they were electrospun using a conventional electrospinning setup. The resulting randomly oriented fibers, as well as control PCL and PVDF fibers, were seeded with adipose tissue-derived MSCs. During cell culture, the cell-seeded scaffolds were regularly exposed to hydrostatic pressure (5 MPa, 0.5 Hz) for one hour. Following each loading session, the scaffolds were transferred to static incubators. Cell attachment and proliferation were observed on the surface of all the scaffolds, with a more significant cell infiltration within the fibrous structure being obtained for the PVDF/PCL fibers. GAG production was more prominent in the piezoelectric scaffolds (PVDF and PVDF/PCL) either with or without hydrostatic pressure. While SOX9 expression, a key chondrogenic regulator, was positively affected by both hydrostatic pressure and piezoelectricity, type II collagen production appeared to be uninfluenced by either parameter. These results appear to corroborate the importance of piezoelectricity on cartilage ECM formation as well as the promising potential of using hydrostatic pressure-type generators for inducing in vitro piezoelectric stimuli to enhance cartilage regeneration [124].

A summary of studies that have described the production of piezoelectric electrospun nanofibers for cartilage TE applications, including a more detailed analysis of the results obtained, is shown in Table 7.

### 4.3. Osteochondral Tissue

Since most studies surrounding the development of piezoelectric scaffolds with potential applications in OCTE have been mostly focused on developing and tunning either AC or SB analogs (with a particular focus on bone), very few publications have described the development of piezoelectric fibrous structures capable of completely replacing or mimicking both the AC and SB regions of OCT.

Khader et al. studied the potential of biodegradable PCL fibers blended with piezoelectric zinc oxide nanoparticles for osteochondral regeneration. PCL/ZnO solutions were first prepared by adding ZnO nanoparticles at different concentrations (1%, 2.5%, 5% and 10%, wt%) to methylene chloride solutions with fixed PCL concentration (13%, *w*/*v*). After homogenization, the solutions were electrospun using a conventional electrospinning setup with a static grounded collector. The authors studied the degradation profile of the fibers. They verified that zinc ions were slowly released over time which, due to their beneficial properties for cartilage and bone regeneration, appeared to corroborate the promising potential for using these biodegradable piezoelectric fibers in OCTE applications. Both PCL and PCL/ZnO fibers were seeded with hMSCs and cultured under static conditions. While a more significant chondrogenic differentiation of hMSCs was observed for the scaffolds with lower percentages of zinc oxide, evidenced by increased type II collagen production as well as the expression of other cartilage-specific genes (e.g., SOX9), the fibers with the highest zinc oxide content (more piezoelectric) promoted the osteogenic differentiation of the hMSCs, evidenced by the augmented alkaline phosphatase activity, type I collagen production, and the expression of other bone-specific genes (e.g., Runx2) [108].

A different study from Damaraju et al. described the development of PVDF-TrFE-based scaffolds with potential applications in OCTE. To produce the random PVDF-TrFE non-woven fibrous mesh, PVDF-TrFE powder (65/35) was blended with methylethylketone (MEK) at a fixed concentration of 25% (wt%). The polymeric solution was then electrospun using a regular setup, and the resulting PVDF-TrFE fibers were annealed in the oven for 96 h at 135 °C, after which the fibers were quenched in ice water for a few seconds. While it was verified that the annealed fibers had a significantly larger relative β phase fraction (75 ± 3.2%) when compared with the non-treated piezoelectric fibers (64 ± 2.8%), it was also reported that the non-treated fibers had better mechanical flexibility compared to the heat-treated fibers. The annealed and non-treated PVDF-TrFE fibers were seeded with hMSCs, and PCL fibers were used as control. Using a dynamic cell culture system in which the scaffolds were regularly compressed (with 10% deformation), the authors were able to verify that as-spun PVDF-TrFE fibers with lower piezoelectric activity-induced mostly chondrogenic differentiation of MSCs. At the same time, heat-treated PVDF-TrFE fibers with higher piezoelectric activity promoted the osteogenic differentiation of MSCs (increased mineralization) [80].

This dual behavior of MSCs related to the piezoelectricity of the fibers described in both previously mentioned studies was considered an interesting potential strategy for developing OCTE constructs using a single cell source. Thus, by creating a piezoelectric gradient scaffold (e.g., poled and non-poled PVDF, a gradient of zinc oxide nanoparticles), it might be possible to direct the proper zone-specific cell differentiation for the different regions within the OCT [9].

In a different approach, Zhang et al. fabricated biphasic OCT scaffolds by combining a piezoelectric PLLA nanofiber layer with a piezoelectric microporous collagen layer. PLLA fibers were first produced by dissolving PLLA (3.5%, *w*/*v*) in chloroform/ethanol (3:1), and afterward, the resulting solutions were electrospun using a conventional electrospinning setup with a static grounded collector. The collagen layer was generated by extracting and purifying type I collagen from pig tendons. The resulting collagen solutions were deposited on the surface of the PLLA fibers, and the scaffolds were then freeze-dried. The biphasic scaffolds were seeded with rabbit bone marrow-derived MSCs and were also applied in vivo, being implanted in artificially generated OCT lesions in rabbit joints. Overall, the generated scaffolds were capable of promoting a robust osteogenic differentiation of MSCs in vitro, as well as accelerating SB emergence and enhancing cartilage formation in vivo. From these results, the authors underscored the promising potential of these biphasic scaffolds for the treatment of large osteochondral defects, which remain an unmet clinical challenge within the field of orthopedics and, as previously discussed, are not adequately addressed with current clinical strategies [125].

An overview of recent research studies using piezoelectric electrospun scaffolds for OCTE applications, including a more detailed analysis of the main results obtained, is presented in Table 7.

## 5. Challenges and Future Perspectives

The main limitation currently restricting a wider application of piezoelectric scaffolds in OCTE applications is the non-biodegradability of most piezoelectric biomaterials, including the piezoceramics, such as barium titanate, as well as the most commonly used PZPs PVDF and PVDF-TrFE. Even though biodegradable PZP choices are available, including PLLA and PHB, they are plagued by their poor mechanical properties and, more importantly, reduced piezoelectric coefficients, which hinder their use in piezoelectricity-based applications [45].

Although biodegradability is considered an important feature for scaffolds, as it allows the material to gradually degrade, giving space for newly formed tissue, an argument can be made for using non-biodegradable materials for regenerating load-bearing tissues with long recoveries (or incapable of full repair) [129]. Indeed, non-biodegradable polymers can support tissue growth for long periods while their mechanical and biological features remain mostly intact, unlike biodegradable materials, which gradually lose their designed properties and therefore become incapable of properly supporting the formation of neo-tissue [129,130]. Still, it is worth noting that the use of these materials for OCTE is not ideal, particularly considering that these scaffolds would most likely have to be removed after a given period to prevent adverse health effects, requiring an additional surgical procedure. Other potential applications for these non-biodegradable piezoelectric fibrous scaffolds may include disease modeling, where they could be used to develop anatomically and physiologically relevant 3D in vitro models of the target OCT and study different osteochondral diseases like OA or rheumatoid arthritis. In addition, non-biodegradable piezoelectric fibrous scaffolds can also be used as coating structures for either bone prosthesis or bone electrodes to improve their integration with the target tissue and, therefore, enhance the long-term performance of these devices [46,81,131,132,133]. A reduced number of strategies have been developed so far to make PVDF and PVDF-TrFE biodegradable. Such solutions involve blending the non-biodegradable PZPs with a highly biodegradable polymeric structure, including cellulose, starch, and PCL [33,134]. However, these processes have yet to be widely applied for producing biodegradable PZPs with high piezoelectric coefficients.

More insight into the processes through which piezoelectric materials interact with biological tissues is still necessary to properly tune the physical, electrical, and biological features of the scaffolds for TE applications and to adequately interpret future and already available data on this topic. Additional research on the piezoelectric properties of the native osteochondral tissue would also be important to not only systemize the collected evidence, which due to different piezoelectricity measurement techniques and animal models used is highly heterogenous, but also identify what would be the ideal target piezoelectric coefficient values for OCTE scaffolds to provide appropriate electrical/mechanical stimuli.

A few technical issues with electrospinning have also been described in the literature. On the one hand, toxic solvents are frequently used for producing electrospinning casting solutions and, therefore, should be completely removed from the final non-woven fibrous mesh to avoid negative effects over cell cultures in vitro or surrounding tissues in vivo [135]. On the other hand, this technique is commonly plagued by jet instability which can be responsible for producing heterogeneous fibers [136]. Some research studies have also pointed out that for small-sized nanofibers, cell migration and tissue integration may be diminished due to the small pore size of the scaffolds [135]. Melt-electrospinning writing (MEW), a development of the conventional technique based on the combination of near-field electrospinning with additive manufacturing methods (e.g., fused deposition modeling), may offer a solution for this limitation since it allows the fabrication of nanofibrous scaffolds with precise control over their pore size and pore interconnectivity [137,138].

Overall, a larger focus on the development of novel piezoelectric bone, cartilage, and OCT fibrous scaffolds would be of interest, given their ability to recapitulate both the structural properties and piezoelectric nature of the tissue’s fibrous ECM as well as of providing a platform for the electrical and mechanical stimulation of cells and tissues, whose role in promoting OCT regeneration has been previously described [9,10,139,140,141]. Another advantage that is very likely to be explored for future OCT regeneration strategies is the possibility of wireless electrical stimulation through the mechanical stimulation (e.g., ultrasound) of piezoelectric scaffolds, which will in part generate electrical signals to modulate cell differentiation. Considering the latest great technological developments, these smart biomaterials may allow future patients to be treated at home by using some brace-like device generating mechanical stimuli to promote the regenerative process of OCT-damaged tissues.

## 6. Conclusions

The development of scaffolds capable of replacing damaged OCT and promoting its regeneration has recently gained renewed attention as a promising alternative for current clinical strategies used to target osteochondral defects and related diseases such as OA. In particular, the production of micro- and nanofibers via electrospinning has attracted interest due to their capacity for recapitulating the fibrous architecture of the OCT’s ECM, facilitating tissue integration as well as providing more native-like biomechanical cues. Although the piezoelectric properties of native OCT have been extensively described in the literature, most OCTE studies tend to focus on recapitulating the main mechanical and structural properties of this load-bearing tissue instead. Additionally, the potential for developing smart electroactive platforms capable of electrically stimulating damaged OCT in vivo to improve its regeneration has already been established.

This review provides an overview of the piezoelectric biomaterials that have been applied thus far for producing piezoelectric fibrous scaffolds via electrospinning for OCTE applications, with examples pertaining to both piezoceramic and piezoelectric polymer classes, including barium titanate, zinc oxide, PVDF, PVDF-TrFE, PHB, and PLLA. While the utilization of piezoceramics in OCTE strategies is limited due to toxicity and environmental concerns surrounding their usage, PZPs have found a wider array of applications, particularly PVDF and its derivatives due to their higher piezoelectric coefficient, flexibility, and biocompatibility. A summary of some of the recent studies focusing on the fabrication and use of piezoelectric fibrous scaffolds for bone, cartilage, and osteochondral TE applications was also presented in this review. Such an eclectic set of studies representing the current knowledge on the topic highlighted the potential of smart piezoelectric nanofibrous scaffolds to improve the current suboptimal TE strategies in OCT regeneration. However, a deeper understanding of the piezoelectric properties of OCT as well as of further optimization of both the scaffold properties and biophysical stimulation protocols are still required to enable the clinical translation of these strategies.

## Figures and Tables

**Figure 1 ijms-23-02907-f001:**
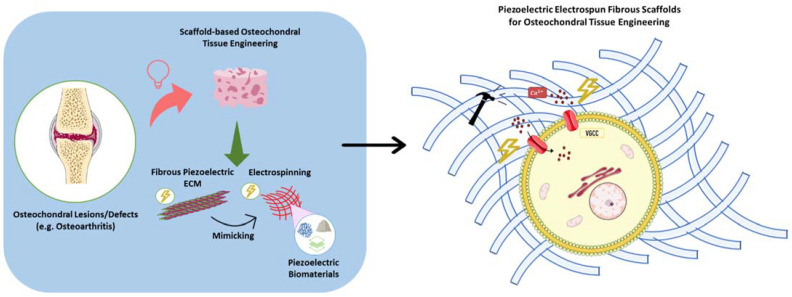
Piezoelectric electrospun fibrous scaffolds for osteochondral tissue engineering applications.

**Figure 2 ijms-23-02907-f002:**
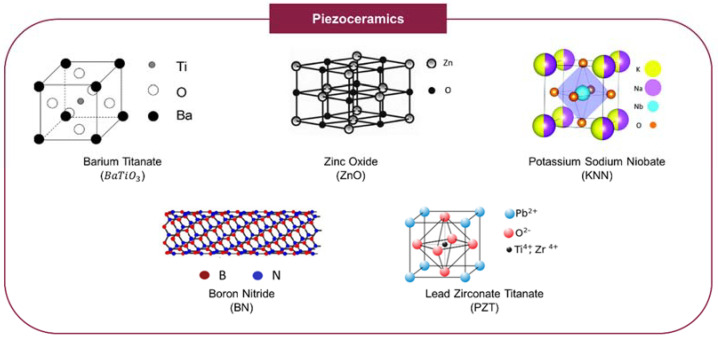
Piezoceramics used for TE applications [51,52,53,54,55]. It should be noted that the chemical structure of piezoceramics is variable, depending mainly on the fabrication process used. Reproduced from Burns and Dolgos [51] with permission from the Centre National de la Recherche Scientifique (CNRS) and The Royal Society of Chemistry. Reprinted from Espitia et al. [52], Copyright 2012, by permission from Springer Nature. Reprinted with permission from Král et al. [53], Copyright 2000 by the American Physical Society.

**Figure 3 ijms-23-02907-f003:**
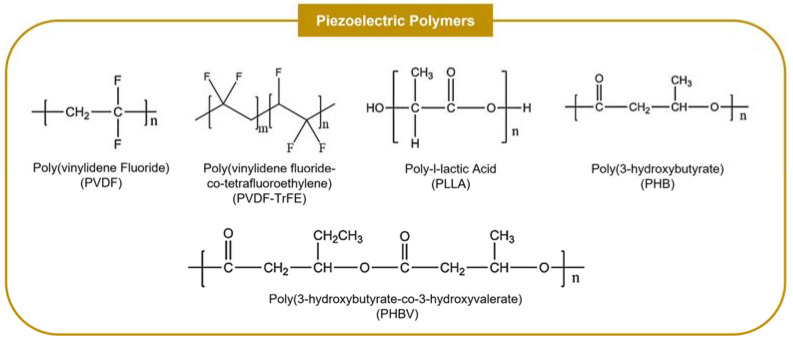
Chemical structures of the main piezoelectric polymers used for TE applications.

**Figure 4 ijms-23-02907-f004:**
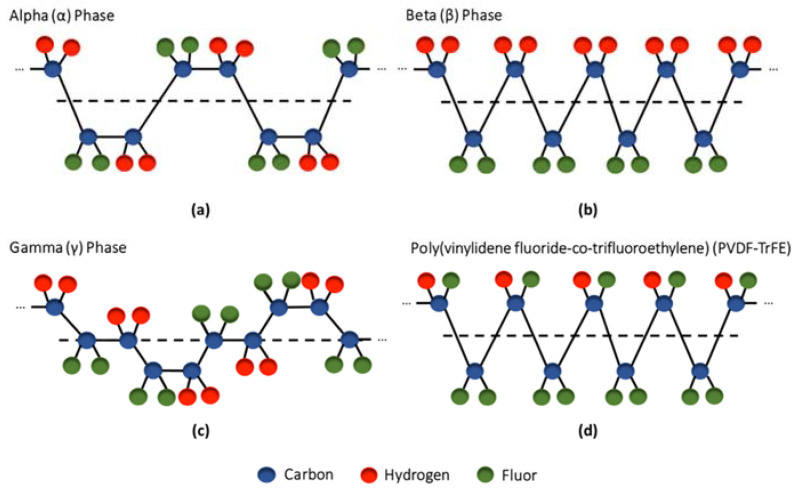
Chemical structures of the most common crystalline phases of PVDF: ((**a**)—α phase, (**b**)—β phase, (**c**)—γ phase) and (**d**) PVDF-TrFE piezoelectric polymers.

**Table 1 ijms-23-02907-t001:** Summary of main dielectric and piezoelectric properties of articular cartilage and cortical and trabecular bone. It should be noted that the values were obtained from different animal models: bovine (*), porcine (*^1^), and human (*^2^). Even though research on the topic is scarce, it is important to consider that the piezoelectric properties of bone and cartilage are likely affected by aging and tissue degeneration, given the effects of these factors on the tissues’ ECM composition (collagen degradation) and mechanical properties that have already been reported in the literature [27,28,29].

	Articular Cartilage	Bone		References
		Cortical	Trabecular	
Conductivity (S/m)	1.14 ± 0.11 *	0.02 *^1^	0.079 *^1^	[30,31,32]
Relative Permittivity	1.39 × 10^3^ *^1^	1.45 × 10^2^ *^1^	2.49 × 10^2^ *^1^	[14]
Piezoelectric Charge Coefficient, d_33_ (pC/N)	0.2–0.7 *^2^	0.7–2.3 *^2^		[33,34]

**Table 2 ijms-23-02907-t002:** Piezoelectric charge coefficient of main piezoceramics applied so far in piezoelectric scaffold-based OCTE strategies.

	Piezoceramics				
	Barium Titanate	HAp	Zinc Oxide	Boron Nitride (BNNTs)	KNN/LKNN
Piezoelectric Charge Coefficient, d_33_ (pC/N)	191	1.5–2.4	12.4	31.2 (d_31_)	KNN: 63 LKNN: 98
References	[42]	[49]	[49]	[56]	[57]

**Table 3 ijms-23-02907-t003:** Summary of main piezoceramics that have been applied so far in piezoelectric scaffold-based TE strategies.

Piezoceramics	Advantages	Disadvantages	Applications	References
Barium Titanate	High piezoelectric coefficient. Biocompatible.	Non-biodegradable. Brittle. Poor thermal stability. Some reports of cytotoxicity.	Bone, Neural, and Skin TE. Theranostics. Drug delivery.	[10,33,42,49]
HAp	Biocompatible. Biodegradable.	Difficult to polarize. Small piezoelectric coefficient (close to bone tissue). Poor mechanical properties.	Bone TE. Implant coating. Filler.	[33,48,49,69]
Zinc Oxide	Biocompatible. Biodegradable. Antibacterial activity.	Cytotoxicity reports (mainly in nanometer-size particles). Small piezoelectric coefficient (compared with other piezoceramics).	Bone and Skin TE. Biosensors. Anti-Cancer Agent. Drug delivery. Theranostics.	[10,33,42,48,49,70]
Boron Nitride (BNNTs)	High piezoelectric coefficient. Biocompatible. High mechanical strength. High surface to volume ratio. Oxidation resistance.	Non-biodegradable. Conflicting reports of cytotoxicity. Negative influence over some cell types.	Bone and Neural TE. Nanotube Internalization (drug delivery). Orthopedic Applications.	[10,42,49,71,72]
KNN/LKNN	High piezoelectric coefficient. Environmentally friendly. Biocompatible. Antibacterial activity.	Reports of cytotoxicity. Difficult processing. Its biodegradability has not been properly investigated.	Bone, Neural, and Skin TE. Drug delivery.	[10,33,42,57]

**Table 4 ijms-23-02907-t004:** Piezoelectric charge coefficient of main piezoelectric polymers applied so far in piezoelectric scaffold-based OCTE strategies.

	Piezoelectric Polymers					
	PVDF/PVDF-TrFE	PLLA	PHB/PHBV	Collagen	Cellulose	Chitosan
Piezoelectric Charge Coefficient, d_33_ (pC/N)	PVDF: 34 PVDF-TrFE: 38	9.82 (d_14_)	PHB: 1.6–2 (d_14_) PHBV: 1.3 (d_14_)	0.2–2 (d_14_)	0.1 (d_31_)	2.54
References	[10]	[42]	[49]	[33]	[76]	[77]

**Table 5 ijms-23-02907-t005:** Summary of the main piezoelectric polymers that have been applied so far in scaffold-based TE strategies.

Piezoelectric Polymers	Advantages	Disadvantages	Applications	References
PVDF/PVDF-TrFE	High piezoelectric coefficient. Biocompatible. Non-cytotoxic. Flexible. Easy to process. High chemical and physical resistance.	Non-biodegradable.	Bone, Cartilage, Cardiac, Neural, and Skin TE. Nerve guidance channels.	[10,33,42,45,48,49,73]
PLLA	Biocompatible. Biodegradable. Non-cytotoxic. Easy to process. Elastomeric behavior. Corrosion resistance.	Low piezoelectric coefficient (compared with PVDF and PVDF-TrFE). Reports of adverse inflammatory reactions.	Medical devices (e.g., screws, fixation rods). Bone, Cartilage, Vascular, Skin, and Neural TE. Drug delivery. Wound dressing.	[1,10,33,42,48,73]
PHB/PHBV	Biocompatible. Biodegradable. Non-cytotoxic. Highly stable. Easy to process.	Low piezoelectric coefficient. Insoluble in water. Poor mechanical properties.	Bone, Cartilage, and Cardiac TE. Medical devices (e.g., sutures, stents). Drug delivery. Theranostics. Wound dressing.	[10,33,42,49,87]
Collagen	Biocompatible. Biodegradable. Non-cytotoxic. Hydrophilic. Low antigenicity.	Low piezoelectric coefficient (when compared to synthetic PZPs). Poor mechanical strength. Toxic crosslinking agents are often used.	Bone, Cartilage, and Skin TE. Drug delivery.	[10,33,42]
Cellulose	Biocompatible. Biodegradable. Non-cytotoxic. High tensile strength. High cell adhesion.	Very low piezoelectric coefficient. Small pore size.	Bone and Neural TE. Drug delivery.	[33,42,49,76]
Chitosan	Biocompatible. Biodegradable. Non-cytotoxic. Antibacterial activity. High porosity.	Low piezoelectric coefficient. Poor mechanical strength.	Bone, Cartilage, and Skin TE. Drug delivery. Anti-Cancer Agent.	[10,33,49,76]

**Table 6 ijms-23-02907-t006:** Examples of piezoceramic or PZP-based electrospun piezoelectric nanofibers described in the literature for bone TE applications.

Fiber Composition	Brief Description	References
Barium Titanate/Calcium Phosphates	Casting Solution: Gelatin (70%, wt%) in distilled water Electrospinning Setup: Grounded roller collector Post-Processing: Crosslinking; Dip coating (calcium phosphates, barium titanate nanoparticles); Annealing Results: Gelatin template structure was completely removed from the fibers after annealing. Different morphological scaffold features were obtained with different post-processing parameters (versatile technique).	[111]
Boron Nitride/Gelatin	Casting Solution: Gelatin (20%, wt%) in acetic acid mixed with boron nitride nanoparticles (0.1%, 1% and 5%, wt%) Electrospinning Setup: Rotating drum collector Post-Processing: Incubation in SBF (mineralization assay) Results: Generated fibers were found to be biodegradable and stable in aqueous environments. The fibers were also capable of supporting the proliferation and osteogenic differentiation of human osteosarcoma cells (without osteogenic factors).	[112]
ZnO-fCNTs/Polyurethane	Casting Solution: Polyurethane (PU, 8%, wt%) in DMF/tetrahydrofuran (THF) (1:1) mixed with ZnO nanoparticles (0.2%, wt%) and carbon nanotubes functionalized with carboxylic groups (fCNTs) (0.1%, 0.2% and 0.4%) Electrospinning Setup: Grounded roller collector Post-Processing: Incubation in SBF (mineralization assay) Results: PU/ZnO scaffolds had improved tensile strength and antibacterial activity. Functionalized fibers were able to improve the proliferation and osteogenic differentiation of pre-osteoblasts.	[118]
PVDF-TrFE (aligned)	Casting Solution: PVDF-TrFE (75/25) (20%, wt%) in DMF/Acetone (3:2) Electrospinning Setup: Grounded roller collector Post-Processing: Annealing; Poling Results: Poled and annealed fibers had the highest relative β phase content and were able to significantly improve the proliferation of mouse pre-osteoblasts.	[113]
PVDF	Casting Solution: PVDF (15%, wt%) in DMF/Acetone (2:3) Electrospinning Setup: Static collector Post-Processing: Oxygen plasma treatment Results: Surface-treated fibers were more hydrophilic than as-spun fibers, and their surface features had long-term stability. More significant cell spreading and integration was observed for surface-treated PVDF fibers.	[114]
PVDF/HAp	Casting Solution: PVDF (16%, wt%) in DMF/Acetone (1:1) mixed with HAp nanoparticles (5% and 10%, wt%) Electrospinning Setup: Rotating drum collector Results: HA-filled PVDF fibers were more hydrophobic and had larger mean diameters and a reduced relative β phase content than simple PVDF fibers.	[115]
PVDF/HAp (coating)	Casting Solution: PVDF (25%, wt%) in DMF/Acetone (3:1) Electrospinning Setup: Static collector Post-Processing: Oxygen plasma treatment; Electrodeposition of HAp (three-electrode cell system) Results: The authors of the study verified the antibacterial effect of the resulting fibers and seeded them with osteoblast-like cells. PVDF/HAp scaffolds improved ALP activity and total protein secretion of the osteoblasts.	[119]
PVDF	Casting Solution: PVDF (22%, wt%) in dimethyl acetamide (DMAC)/Acetone (1:1) Electrospinning Setup: Static collector. Note: two voltages with different polarities were used to produce PDVF fibers with different surface potentials (PVDF (+), PVDF (−)) Results: PVDF (−) fibers had the highest surface potential (similar to the potential of osteoblast-like cells). More significant cell proliferation was observed for PVDF (−) fibers, which were also found to accelerate collagen mineralization.	[120]
PVDF- Barium Titanate/PVA (*coaxial*)	Casting Solution: Core—PVDF (27%, wt%) in dimethyl sulfoxide (DMSO)/Acetone (3:2) mixed with barium titanate nanoparticles (1%, 2% and 5%, wt%); Sheath—PVA (15%, wt%) in DMSO/Ethanol (9:1) Electrospinning Setup: Coaxial setup Post-Processing: Incubation in SBF (mineralization assay) Results: The addition of PVA improved the wettability and biodegradability of the coaxial fibers. Barium titanate, in turn, enhanced the bioactivity and mechanical properties of the scaffolds. The generated scaffolds were able to promote the osteogenic differentiation of MSCs.	[121]
PHBV/SiHAp	Casting Solution: PHBV (23%, wt%) in chloroform mixed with SiHAp nanoparticles (10%, wt%) Electrospinning Setup: Rotating drum collector Results: HAp contributed to a slight increase in the piezoelectric coefficient of the resulting fibers. Enhanced MSC proliferation and osteogenic differentiation results were observed for the functionalized scaffolds (higher calcium accumulation).	[116]
PLLA/Barium Titanate (random and aligned)	Casting Solution: PLLA in trifluoroethanol mixed with barium titanate nanoparticles (1%, 3%, 5%, 7% and 10%, wt%) Electrospinning Setup: Static collector (random fibers); Rotating drum collector (aligned fibers) Results: The addition of barium titanate improved the surface roughness and wettability of the fibers. MSCs seeded on the random functionalized scaffolds displayed improved osteogenic differentiation.	[117]

**Table 7 ijms-23-02907-t007:** Examples of piezoceramic or PZP-based electrospun piezoelectric nanofibers described in the literature for articular cartilage (1) and osteochondral (2) TE applications.

Fiber Composition	Brief Description	References
PHBV/Barium Titanate (1)	Casting Solution: PHBV (15% and 20%, wt%) in chloroform/methanol (3:2) mixed with barium titanate nanoparticles (5%, 10% and 20%, wt%) Electrospinning Setup: Static collector Post-Processing: Poling Results: The increase in barium titanate concentration resulted in an increase in the Young’s moduli of the resulting scaffolds and an increase in their piezoelectric coefficients. Enhanced chondrocyte adhesion, proliferation, and expression of chondrogenic markers were observed for the fibers containing barium titanate. Poled fibers exhibited increased cell proliferation and collagen production.	[123]
PHB/Chitosan (1)	Casting Solution: PHB (9%, wt%) in trifluoroacetic acid (TFA) mixed with chitosan (5%, 10%, 15% and 20%, wt%) Electrospinning Setup: Static collector Results: Due to the addition of chitosan, the scaffolds exhibited improved hydrophilicity and biodegradability, as well as reduced porosity and Young’s modulus. The fibers were seeded with isolated rabbit chondrocytes to evaluate cell adhesion. Improved cell spreading and attachment were observed for the scaffolds with chitosan.	[109]
PHB/CNT/Chitosan Hyaluronic Acid (HA) (1)	Casting Solution: PHB (9%, wt%), chitosan (20%, wt%) and carbon nanotubes (CNTs) (1%, wt%) in TFA mixed with hyaluronic acid (HA) (5%, 10% and 15%, wt%) Electrospinning Setup: Static collector Post-Processing: Incubation in SBF (mineralization assay) Results: While chitosan was added to improve the biodegradability and hydrophilicity of the scaffolds, CNTs were used to enhance their mechanical properties. The addition of HA contributed to an increase in the wettability of the resulting scaffolds and a slight reduction of their Young’s modulus and porosity. Chondrocytes were able to attach and proliferate on the surface of these bioactive fibers corroborating their biocompatibility.	[126]
PVDF/PCL (1)	Casting Solution: PCL in DMF/THF (1:1) mixed with PVDF in DMAC/Acetone (1:1) (50:50) Electrospinning Setup: Static collector Results: The resulting fibers were seeded with adipose tissue-derived MSCs and were regularly exposed to hydrostatic pressure. While the piezoelectricity of PVDF was found to promote GAG production and SOX9 gene expression, as well as contribute to improved cell proliferation and integration, the hydrostatic stimuli promoted the production of aggrecan. Type II collagen expression appeared to be uninfluenced by either the presence of PVDF or the applied mechanical stimuli.	[124]
PLLA/PCL (1)	Casting Solution: PLLA (12% and 20%, wt%) and PCL (12% and 20%, wt%) in dichloromethane (DCM)/DMF (75:25). Note: Two polymeric concentrations were considered to obtain fibers with different diameters (800 nm and 1.8 μm) Electrospinning Setup: Static collector Results: PLLA, PCL, and PLLA/PCL scaffolds were seeded with MSCs. High levels of cell proliferation were registered for all conditions, particularly for the PCL and composite fibers. All scaffolds could promote the chondrogenic differentiation of MSCs in the absence of chondrogenic factors, with a more significant expression of type II collagen for the PLLA-containing fibers. Improved cell adhesion and chondrogenic differentiation were reported for the fibers with larger diameters.	[127]
ZnO/PCL (2)	Casting Solution: PCL (13%, wt%) in methylene chloride mixed with ZnO nanoparticles (1%, 2.5%, 5% and 10%, wt%) Electrospinning Setup: Static collector Results: Zinc ions were slowly released over time with the degradation of PCL (relevant regenerative potential). The piezoelectric scaffolds were seeded with MSCs. For higher ZnO concentrations (more piezoelectric), osteogenic differentiation was promoted, for the fibers with lower ZnO concentrations (less piezoelectric) chondrogenic differentiation of MSCs was more prominent.	[108]
PVDF-TrFE (2)	Casting Solution: PVDF-TrFE (65/35) (25%, wt%) in MEK Electrospinning Setup: Static collector Post-Processing: Annealing Results: Annealed fibers had a larger relative β phase content but also worse mechanical flexibility compared with as-spun PVDF-TrFE fibers. After being seeded with hMSCs, it was verified that while the annealed fibers (more piezoelectric) promoted the osteogenic differentiation of MSCs, the as-spun fibers (less piezoelectric) promoted MSCs chondrogenic differentiation.	[80]
PLLA/Collagen (biphasic) (2)	Casting Solution: PLLA (3.5%, wt%) in chloroform/ethanol (3:1) Electrospinning Setup: Static collector Results: Type I collagen was extracted from pig tendons and freeze-dried on top of the generated PLLA piezoelectric fibers. While in an in vitro setting, the scaffolds were able to promote the osteogenic differentiation of seeded MSCs. When implanted in vivo in damaged rabbits’ OCT, the biphasic scaffolds enhanced AC formation and accelerated SB emergence.	[125]
P(LLA-CL)/Collagen/Hyaluronan (chondral region) β-TCP (osseous region) (biphasic) (2)	Casting Solution: P(LLA-CL) (75/25) and type I collagen (8%, wt%) in hexafluoroisopropanol (HFIP) Electrospinning Setup: Dynamic liquid electrospinning setup Results: P(LLA-CL) (blend of PLLA and PCL) piezoelectric fibers were used as a template to produce analogs for the AC region of the OCT: a type I collagen and hyaluronan blend was deposited on the surface of the fibers, and the scaffolds were freeze-dried (Yarn-CH). β-Tricalcium phosphate (TCP) piezoelectric microporous structures were developed using a high-temperature melting method for replacing the osseous region of the OCT. Prior to being implanted in vivo, the Yarn-CH and TCP regions were attached using collagen and hyaluronan as cement, after which the OCT scaffolds were frozen and BM-MSCs were expanded on the surface of the biphasic scaffolds. Successful repair of OCT defects in rabbits (with improved quality and speed) was reported.	[128]

## Data Availability

Not applicable.

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
