# Peer review of "Piezoelectric Electrospun Fibrous Scaffolds for Bone, Articular Cartilage and Osteochondral Tissue Engineering"

_ijms, 2022, doi:10.3390/ijms23062907_

Round 1
Reviewer 1 Report
- Lines 37-39: You speak about direct and reverse piezoelectric effects, and at lines 11-12 you provide an equation (1) for coefficients relating the electric field strength and mechanical stress. I think it shall be pointed more clearly, which terms in Eq.(1) correspond to direct and respectively to the reverse piezoelectric effects.
- Lines 56-57: You mention all four piezoelectric coefficients but you present an equation only for one of them. How these coefficients are related to each other and, in the context of review, why dij coefficients are emphasised as being computed?
- In my opinion the reported piezoelectric coefficients for the various biological materials could be arranged in a Table for a faster analysis, but this is up to Authors.
- In Abstract you mention "While the piezoelectric properties of the OCT have been extensively reported in different studies, they keep being neglected in the design of novel OCT scaffolds, which tend to focus primarily on the tissue’s structural and mechanical properties". Of course the mechanical and structural properties of scaffolds are also very important, especially when we seek to replace damaged OCT and promoting regeneration. Thus, it would very useful if you could provide additional details about these physical properties for the materials discussed in this work.
Author Response
We would like to acknowledge reviewer 1 comments and suggestions, which we believe were crucial to improve the quality of our manuscript.
Therefore, we addressed all the reviewers’ comments in our point-by-point response document (PDF attached) and made the corresponding changes in the manuscript (highlighted) as recommended.

Reviewer 2 Report
General comments
The manuscript concerns the ongoing research of piezoelectric materials for OCT regeneration. The manuscript is sufficiently organized.
A paragraph that describes the mechanical properties of the electrospun piezoelectric scaffolds is required to highlight the potential application of these classes of materials for OCTE.
Detailed comments
- line 85: you have to detail the piezoelectric properties of OCT and the cells/molecules that act for this behavior
- line 114: you have to detail if the scaffolds you are referring to, have piezoelectric properties
- line 120: clarify if you are referring to in vitro models (physiologically relevant platforms)
- line 1125: better explain to what you are referring (related applications)
- line 147: “piezoelectric” --> you have to quantify the electrical properties and the piezoelectric one
- a table summarizing the electrical and piezoelectric properties of AC and bone is required. You have to clarify if these properties are affected by age, pathologies, …
- line 202: specify if piezoceramics are used for OCTE scaffolds or you are describing, in general, the possible use of piezoceramics in TE
- line 229: “when mechanical stress …” --> you have to specify how in vivo the stimulus is applied to the piezoelectric ceramic material
- line 245: better explain if you mean that HAp is per sè piezoelectric or you have to synthesize adequately the HAp formulation
- lines 282-284: deeper detail this point
- Table 1: you have to add a column reporting the possible applications for each of the piezoceramics you listed in the table; as your manuscript is focused on OCT, it is expected to have an overview of the possible applications in the area of interest
- polymeric piezoelectric materials: see comments for piezoceramics
- line 293: it seems that only one natural polymer shows piezoelectric properties. Is it correct? are you sure?
- line 464: better explain the technologies used for the production of the scaffolds for OCT regeneration and explain how electrospinning can be a real alternative for the realization of a 3D scaffold for OCT
Author Response
We would like to acknowledge reviewer 2 comments and suggestions, which we believe were crucial to improve the quality of our manuscript.
Therefore, we addressed all the reviewers’ comments in our point-by-point response document (PDF attached) and made the corresponding changes in the manuscript as recommended.

Round 2
Reviewer 1 Report
The Authors took into account my recommendations.